# Presynaptic Acetylcholine Receptors Modulate the Time Course of Action Potential-Evoked Acetylcholine Quanta Secretion at Neuromuscular Junctions

**DOI:** 10.3390/biomedicines10081771

**Published:** 2022-07-22

**Authors:** Ellya A. Bukharaeva, Andrey I. Skorinkin, Dmitry V. Samigullin, Alexey M. Petrov

**Affiliations:** 1Kazan Institute of Biochemistry and Biophysics, Kazan Federal Scientific Centre “Kazan Scientific Centre of Russian Academy of Sciences”, 2/31 Lobatchevsky Street, 420111 Kazan, Russia; askorink@yandex.ru (A.I.S.); samid75@mail.ru (D.V.S.); apneurosci@gmail.com (A.M.P.); 2Department of Radiophotonics and Microwave Technologies, Kazan National Research Technical University named after A.N. Tupolev, 420111 Kazan, Russia; 3Institute of Neuroscience, Kazan State Medical University, 49 Butlerova Street, 420012 Kazan, Russia

**Keywords:** neuromuscular junction, evoked quantal acetylcholine release, kinetics of the quantal secretion, presynaptic acetylcholine receptors

## Abstract

For effective transmission of excitation in neuromuscular junctions, the postsynaptic response amplitude must exceed a critical level of depolarization to trigger action potential spreading along the muscle-fiber membrane. At the presynaptic level, the end-plate potential amplitude depends not only on the acetylcholine quanta number released from the nerve terminals in response to the nerve impulse but also on a degree of synchronicity of quanta releases. The time course of stimulus-phasic synchronous quanta secretion is modulated by many extra- and intracellular factors. One of the pathways to regulate the neurosecretion kinetics of acetylcholine quanta is an activation of presynaptic autoreceptors. This review discusses the contribution of acetylcholine presynaptic receptors to the control of the kinetics of evoked acetylcholine release from nerve terminals at the neuromuscular junctions. The timing characteristics of neurotransmitter release is nowadays considered an essential factor determining the plasticity and efficacy of synaptic transmission.

## 1. The Time Course or Kinetics of the Evoked Acetylcholine Quanta Release

Following the activation of motoneuron, acetylcholine (ACh) released from the presynaptic nerve terminals initiates a depolarization of the muscle end plate, which triggers an action potential causing muscle-fiber contraction. Vesicular or “quantal” hypothesis of ACh release introduced by B. Katz presumes that the nerve impulse evokes the release of tens (warm-blooded) to hundreds (cold-blooded) of ACh quanta from the motor nerve terminals at the neuromuscular junction [1,2]. This process is called a phasic neurotransmitter secretion. Its nature is probabilistic, and therefore it cannot be absolutely synchronous. The individual ACh quanta are released with a synaptic delay. This is estimated using extracellular recording as the interval between the peak of the presynaptic action potential and the onset of uniquantal postsynaptic response under low calcium and high magnesium levels in extracellular solution [3,4]. The intervals estimated in this way are called “real” synaptic delays. This term makes it possible to distinguish synaptic delay from the latency, which denotes an interval between the stimulus (not nerve action potential) and the postsynaptic response onset [3]. At room temperature, ACh quanta release usually begins 0.2–0.5 ms after the peak of action potential (a minimal synaptic delay), and then decays at between 4–7 milliseconds (Figure 1A,B) [3,4,5,6,7]. Analysis of the delay distribution of individual quantal events gives insight into the intraterminal processes determining the time course of release probability [5,8,9,10]. The timing of neurotransmitter release can be modulated by cholinergic, adrenergic, and purinergic receptors, as well as altered in some mouse models of motor and neurodegenerative diseases, which we will describe in the Discussion section.

Computational analysis of the synaptic delay distribution (at low external calcium) at the frog neuromuscular junction showed that it can be represented as the sum of normal and exponential distributions, and this suggests a two-stage mechanism of early and late periods of phasic synchronous secretion [11]. Initially the idea about an existence of such variety at the time of ACh release was expressed by Barrett and Stevens [5] and Goda and Stevens [12].

The time course of ACh quanta release depends differently on the motor nerve stimulation pattern and on calcium ion entry into the nerve terminals. Under physiological calcium levels, ACh quanta, which form multiquantal postsynaptic response, are also released with different latencies. Their distribution is obtained by different computational methods from multiquantal end-plate currents and spontaneous miniature end-plate currents [13,14,15,16]. Analysis of latency distribution indicates that these ACh quanta are also not released simultaneously (Figure 2A,B) and the two components can be distinguished.

## 2. The Role of Evoked Quantal-Secretion Kinetics in the Formation of a Multiquantal Postsynaptic Response

Before the consideration of the role of presynaptic ACh receptors (AChRs) in the modulation of the time course of quantal release, we would like to point out the significance of the evoked ACh release kinetics for neuromuscular transmission. Certainly, the main factor defining the efficiency and reliability of the neurotransmission is the number of ACh quanta released in response to a nerve action potential, i.e., quantal content. This parameter determinates the peak amplitude of the multiquantal end-plate potential (EPP) and if the amplitude exceeds a critical level of depolarization, then the EPP induces the muscle-fiber action potential, and eventually the muscle-fiber contraction occurs. In the late 1960s, Stevens introduced the stochastic approach for the kinetics analysis of quanta released following an impulse at motor nerves [17]. In the subsequent decades, it was developed and shown that the release kinetics of the individual quanta forming a multiquantal response is significant for both the amplitude and rise time of the postsynaptic response [8,18,19,20,21,22]. This can be demonstrated in a simple scheme (Figure 1C). The individual ACh quanta are released with different delays (Figure 1A). If quanta were released absolutely synchronously (Figure 1(Ca)), then the peak amplitude of the end-plate current would be almost equal to the sum of the amplitudes of uniquantal responses and its rise time would be the same as the rise time of the uniquantal responses (Figure 1(Cb)). When quanta are released nonsimultaneously with variable delays (Figure 1(Cc)), the total response becomes smaller in the peak amplitude and has a longer rise time (Figure 1(Cd)). The peak amplitude of such “asynchronous” signal does not reach a critical level of depolarization and it cannot cause the action potential of the muscle fiber (Figure 1D). The convolution method with real parameters of the ACh quanta-release dispersion and uniquantal responses has shown that when the dispersion time of individual ACh quanta release increases, then the EPPs’ amplitude can decrease by from 17% to 40% and their rise times become more prolonged by 20–50% [11,19,21,22]. Thus, the kinetics of ACh quanta release has significance for neurotransmission efficacy and can be changed under various physiological and pathological conditions. Herein, we discuss the ability of presynaptic AChRs to modulate the kinetics of ACh quanta release at the neuromuscular junctions.

## 3. Presynaptic Acetylcholine Receptors Modulate the Kinetics of Acetylcholine Quanta Release

Neuromuscular synapses of vertebrates contain presynaptic AchRs, which implement a modulatory action of ACh on its own release. Despite a long history of studies showing the effects of both AChR activators and blockers on quantal ACh secretion [23,24,25,26,27,28], there is still no consensus on the role of presynaptic ionotropic nicotinic (nAChR) and metabotropic muscarinic (mAChRs) receptors at the motor-nerve terminals in modulation of the kinetics of evoked ACh release. The data about the effects of nAChR blocker d-tubocurarine [29] and muscarinic agonists on the synaptic delays in the neuromuscular junctions [30,31,32,33,34,35,36] suggest the participation of AChRs in regulation of the time course of ACh release.

Our investigations have shown that exogenous ACh and its nonhydrolyzable analog carbacholine affect not only the quantal content but also the synaptic delay dispersion of the uniquantal end plate currents (EPCs) at the frog neuromuscular junction [34,35]. Both these drugs changed the time course of quanta secretion in distal parts of long frog nerve terminals. ACh and carbacholine caused an increase in minimal synaptic delay, a shift in the delay-distribution histogram to larger values, and a rise in variability of synaptic delays (Figure 3A–C), i.e., augmentation of ACh quanta secretion asynchrony [34,35]. The increasing dispersion of the quanta release resulted in a decrease by ~17% in the multiquantal EPC amplitude reconstructed with the convolution method, taking into account the changed synaptic delay distribution (Figure 3D).

Under physiological conditions, neuromuscular junction operates in a rhythmic activity mode with transmission of frequency from 10 to 100 Hz. Endogenous ACh released from nerve terminals can affect presynaptic AChRs, causing changes in kinetics of ACh quanta secretion. Our study on the kinetics of ACh quanta release during rhythmic stimulation in frog synapses showed that the motor-nerve stimulation at 10–100 Hz increased the rise time of the EPCs [37]. The high-frequency stimulation (50 Hz) produced a delay in the onset of ACh quanta release by ~24% compared to the mean value in control. This might be a result of both a slowed propagation of the nerve impulse and an increased dispersion of the synaptic delays. Analysis of the release kinetics of ACh quanta forming multiquantal EPCs showed an increase in asynchrony of the time moments of quanta release, which can be due to the effect of endogenous ACh. Therefore, a decrease in the multiquantal EPC amplitude after AChR activation occurred due to both the quantal content drop and the increase in asynchrony degree of ACh quanta release. These data suggest that the high-frequency activity of the motoneurons is indeed accompanied by a modification of the kinetics of ACh quanta release from the nerve terminals. The desynchronization of exocytosis events at individual active zones may account for a small prolongation of the EPC rise time at the onset of a high-frequency train (50–100 Hz). It could be proposed that near physiological conditions (Ca^2+^ concentration, a typical for fast motor units’ high-frequency firing), a modification of the ACh time-course release might be one of the general presynaptic mechanisms that contributes to synaptic plasticity during motor-neuron activity *in vivo*. Since ACh and carbacholine can activate both nAChRs and mAChRs, pharmacological analysis using specific agonists and antagonists is necessary to determine the type of cholinergic receptors that modulate the kinetics of neurosecretion.

### 3.1. Do Nicotinic Agonists Change the Kinetics of Evoked ACh Neurosecretion?

There are a large amount of data about the effects of nAChRs agonists and antagonists on the ACh quanta number released in response to nerve stimulus [23,25,26]. Nevertheless, the results of the investigations about changes in the kinetics of ACh quanta release under specific nAChR modulation are scarce. A specific agonist of nAChRs nicotine at the micromolar concentrations significantly increased a minimal synaptic delay as well as the main modal value of the synaptic delay histogram in the distal region of the frog nerve terminal; in addition, a blocker of nAChRs d-tubocurarine prevented the desynchronizing action of both ACh and carbacholine [35]. However, d-tubocurarine alone did not change the synaptic delays. This is different from the results obtained by Matzner et al. [29], where d-tubocurarine shifted the peak of the histograms to longer delays and prolonged the minimal delay. This discrepancy may be due to the various experimental conditions. Available data indicate that the effects of nAChR activation can change the neurotransmitter release, affecting both the number of ACh quanta released and the synchronicity of the evoked exocytotic events.

### 3.2. Muscarinic Agonists and Antagonists Modulate the Time Course of Evoked ACh Secretion

Much more information is available about the contribution of presynaptic muscarinic receptors that modulate the evoked secretion of ACh quanta [27,28]. Different types (M1, M2, M4) of mAChRs can take part in the evoked transmitter-release modulation via both positive and negative feedbacks [38,39,40]. Specifically, M1 AChRs are responsible for the facilitation of synaptic transmission, whereas M2 receptor activation causes the depression [31,39,40].

Immunohistochemical staining with specific antibodies enables the detection of distinct receptor subtypes in synaptic regions. Santafe et al. revealed the immunoexpression of all five (M1-M5) currently identified mAChR subtypes in the rat neuromuscular junction [32,33]. Tsentsevitsky et al. then identified all these subtypes of mAChRs at the motor-nerve terminals of frog *cutaneous pectoris* muscle [41]. The diversity of mAChRs could be one of the reasons for dependence on the observed effects of their modulation on the experimental conditions and might explain the contradictory data on the participation of different mAChRs in the regulation of synaptic transmission.

Activation of mAChRs by muscarine, which is an agonist for all mAChRs, induced the shortening distribution of “real” synaptic delays of EPCs recorded in a low Ca^2+^/high Mg^2+^ solution at the frog neuromuscular junction. Selective antagonists of M1, M2, and M4 mAChRs have prevented the synchronizing action of muscarine on ACh secretion [41].

There are also data on the influence of mAChR inactivation on the kinetics of ACh secretion. Slutsky et al. showed that methoctramine, a selective M2/M4 antagonist of mAChRs, caused longer synaptic delays under a low temperature and slowed down the rise time of the EPCs at the frog neuromuscular junction [42,43]. However, in our experiments under normal temperature [35] muscarinic agonists did not change the time course of ACh quanta release (Table 1). The effects of muscarinic agonists and antagonists on ACh release from the frog motor-nerve terminals depended strongly on the amplitude of local depolarizing pulses, temperature, and the conditions of Ca^2+^ ions’ input into the nerve terminals [30,39,43].

At a later date, it was shown that muscarine decreased the quantal content of the multiquantal EPCs and synchronized the quanta secretion [41]. The depressing action of muscarine on the EPC quantal content was abolished by pretreatment with the specific blockers of muscarinic receptors, namely 4-DAMP and J 104129 (Table 1). The application of M2 blocker AF-DX 116 itself resulted in the shortening of the time interval during which 85% of the release events occur. This finding may be related to both the inherent effect of this compound and the ability of ACh to desynchronize the secretory process specifically via M2-receptors [41].

These data are different from I. Slutsky’s observation that the blockade of mAChRs led to desynchronization of the release of ACh quanta at the rodent’s neuromuscular junctions [42]. In the knockout mice lacking presynaptic M2-type receptors, ACh secretion from the motor-nerve terminal starts sooner and lasts longer than in wild-type mice. Thus, the presynaptic M2 type of mAChRs participates in the control of the time course of evoked ACh release at the mammalian neuromuscular junctions.

## 4. Possible Mechanisms for Modulation of the Kinetics of Quanta ACh Secretion by Presynaptic AChRs

The molecular mechanisms for regulation of quantal release time course are not fully elucidated. The ACh release synchronicity depends on the state of the vesicle exocytosis machinery. The critical factors for quanta release are the accessibility of synaptic vesicles from the ready releasable pool, the activity of exocytosis proteins, and the proximity of the synaptic vesicle to the active zone, where vesicular protein synaptotagmin I/II, fast Ca^2+^ sensor, is located near to presynaptic voltage-gated Ca^2+^ channels [44,45].

Based on the current experimental data, the several most probable stages of mechanisms controlling synaptic-vesicle exocytosis can be identified. These steps, individually or in combination, may be involved in the regulation of the kinetics of evoked ACh secretion:

(i) The synchronization of neurosecretion from motor-nerve terminals depends on Ca^2+^ entry into the axoplasm. It is known that an increase in the concentration of Ca^2+^ in the nerve terminal initiates a fusion of synaptic vesicles with the presynaptic membrane. Therefore, a higher Ca^2+^ should accelerate the rate of the release reaction and result in short synaptic delays [46,47,48]. The ACh quanta-evoked secretion synchronicity is affected by the concentration of Ca^2+^ ions both in the extracellular environment and in the axoplasm [46,47,49]. Other factors, such as the rhythmic activity of the synapse [37], temperature [5,50], as well as the activity of presynaptic receptors and exocytosis proteins [51], can influence the synchrony of evoked release via Ca^2+^-dependent pathways. A high level of the ACh-release asynchronicity was observed at the developing neuromuscular junctions [52], characterized by different set of voltage-gated Ca^2+^ channels, which trigger exocytosis. Mathematical analysis of quanta-release kinetics showed that under low Ca^2+^ input and fixed endogenous Ca^2+^ buffers (i.e., buffers having a diffusion rate much lower than the diffusion rate of free Ca^2+^), a more asynchronous ACh quanta release should occur. If in this system a mobile Ca^2+^ buffer (with a high diffusion capacity) is added, the dispersion of synaptic delays decreases and ACh release becomes more synchronous [22]. This corresponds to the changes in secretion time-course parameters that were found experimentally [47,49]. The spatiotemporal Ca^2+^ profile near the active zone can change the synaptic delay dispersion in the dependence on either extracellular Ca^2+^ concentration or intracellular Ca^2+^ buffer activity [45]. The augmentation of the Ca^2+^ entry leads to the rise in the peak rate of vesicle exocytosis and ACh release becomes more synchronous [48]. Therefore, the change in Ca^2+^ dynamics within the active-zone regions is the most potent mechanism for regulation of the synaptic delays and quanta secretion synchronicity. It is important to note that the presynaptic AChRs activation induce intracellular Ca^2+^ modulation [53].

(ii) Nicotinic AChRs are ligand-gated cation channels. Following the activation of the presynaptic nAChRs, Ca^2+^ concentration in the vicinity of the active zone can be changed by several mechanisms: (a) direct Ca^2+^ influx through the nAChRs; (b) Ca^2+^ input via voltage-gated Ca^2+^ channels that can be activated due to the nAChR-mediated depolarization of the presynaptic membrane; (c) Ca^2+^-induced Ca^2+^ release from the endoplasmic reticulum [54]. According to the dependence of the secretion kinetics on the concentration of Ca^2+^ described above, one would expect an increase in synchronicity of the ACh quanta secretion upon nicotinic AChR activation. However, we have observed an increase in the dispersion of synaptic delays under nicotine treatment [34].

As an alternative option for the nAChR participation in the regulation of the secretion kinetics, we have to take into account that nAChRs can be differentially coupled to Ca^2+^-induced Ca^2+^ release and/or voltage-gated Ca^2+^ channels. Ca^2+^ input through nAChRs can positively modulate the activity of L-type Ca^2+^ channels in the neuromuscular junctions [26,55]. This type of Ca^2+^ channel is responsible for a higher degree of secretion asynchrony in the neuromuscular synapses of newborn rats [52,56] as well as at adult frog neuromuscular junctions [57]. Ca^2+^ influx through L-type channels can regulate a number of signaling enzymes as well as ion channels, which finally affects the state of exocytotic machinery.

One is clear that synaptic delay is dependent on the interaction between Ca^2+^ entering through both the ion channels of the AChRs and voltage-gated Ca^2+^ channels located near to the AChRs. The computer simulation showed that an increase in the distance between the voltage-gated Ca^2+^ channels and synaptic vesicles or the removal of a significant number of Ca^2+^ channels from the active zone led to a less pronounced increase in synaptic delay, i.e., to a lesser degree of asynchrony of the ACh secretion. Moreover, the synaptic delay is sensitive to alterations in the spatiotemporal Ca^2+^ dynamics near the active zone at the time of release. These alterations can be caused by manipulations of the density and localization of voltage-gated Ca^2+^ channels or the concentration of Ca^2+^ buffering proteins [45] as well as the external ionic composition [47]. Thus, different mechanisms may be considered that implement the effects of presynaptic nAChR activation on the time course of ACh secretion.

It should be noted that there is a relatively new hypothesis about the participation of postsynaptic nAChRs in modulating both the quantal content and time course of ACh secretion. This mechanism is related to homeostatic synaptic plasticity [58]. A partial block of the postsynaptic AChRs by d-tubocurarine induced a fast potentiation of ACh secretion and increased synchronicity of quanta release. The changes in the release kinetics following the application of d-tubocurarine were distinct from those induced by either increasing the external Ca^2+^ level or widening the presynaptic action potential [59].

(iii) mAChRs are metabotropic, G protein-coupled receptors. As we described above, their subtypes regulate both the quantal content and the neurotransmitter release kinetics. mAChRs can interact with the exocytotic machinery, which includes the proteins implicated in the control of fast Ca^2+^-regulated exocytosis. In turn, SNARE proteins, synaptotagmin, and the Ca^2+^ channels can be involved in the regulation of ACh release kinetics [60,61].

M2 and M4 subtypes of mAChRs are coupled to the Gi/o proteins and inhibit an activity of adenylyl cyclase. Early, it was shown that an increase in intracellular cAMP concentration results in the synchronization of ACh quanta release [62]. M2 and M4 receptors also activate G-protein-gated rectifying potassium (GIRK) channels, which lead to hyperpolarization of the membrane in excitable cells. The M2-type of mAChRs controls the activity of GIRK channels and causes a desynchronization of the neurotransmitter release at low external Ca^2+^ [63]. mAChRs can change both Ca^2+^ influx and activity of intracellular enzymes, particularly PKC and PKA [33,64]. In neurons, M2 and M4 types mAChRs are present on the axon terminals and inhibit neuronal excitability, which results in the negative feedback regulation of ACh release [65]. Activation of the M2 subtype of AChRs also can inhibit PKA by downregulating the Cβ subunit of PKA. This caused a decrease in PKA-dependent phosphorylation of SNAP-25, a protein from the SNARE complex [66].

A detailed investigation by J. Tomas’s laboratory showed that the M1 and M2 types of mAChRs can induce activation of PDK1, a master kinase that may control other kinases. M1-receptor stimulation increased the interaction of PKCs with the presynaptic membrane and promoted phosphorylation of Munc18-1, SNAP-25, and MARCKS. In contrast, M2 downregulated PKCε through a PKA-dependent pathway, which caused the inhibition of Munc18-1 synthesis and PKC phosphorylation. These results show the existence of a balance between mAChRs, which can regulate the presynaptic PKC and their action on ACh release through the SNARE-SM mechanism [67] or L-type Ca^2+^ channels–PKC ensemble [63]. Thus, there are complex mAChR-dependent mechanisms that can change not only a number of ACh quanta released but modulate the time course of the secretion.

The accumulated data indicate that the changes in kinetics of ACh release from the motor-nerve terminals may be mediated by the various types of presynaptic AChRs and complex downstream signaling mechanisms. The kinetics of the changes in quanta release during in vivo activity should be taken into account as a pathway underlying the plasticity in chemical synapses. In addition, the contribution of presynaptic autoreceptors for ACh to shaping the time course of neurosecretion can serve as an additional feedback mechanism for adaptation of neurotransmitter release to the pattern of the nerve activity.

Changes in the time course of phasic secretion were observed in a wide range of pathological conditions. A marked enhancement of the asynchronous ACh release in combination with decreased amplitude of the EPPs was found in the neuromuscular junctions of model mice with spinal muscular atrophy [68]. Similarly, in patients with spinal muscular atrophy characterized by severe muscle weakness, the amplitude of the postsynaptic response is reduced by 50%, and their rise time is increased by more than twice, due to the increased asynchronous release of ACh quanta [69]. Immunoglobulins G from sporadic ALS patients selectively bind to presynaptic nerve terminals of motoneurons and potentiate significantly asynchronous component of evoked ACh release. A neurotoxic heavy metal cadmium (at nanomolar concentrations) by producing mitochondrial-dependent oxidative stress desynchronized ACh release at the frog neuromuscular junctions [70]. Note that the Cd^2+^-induced desynchronization as well as oxidative burst can be greatly augmented by Zn^2+^, which can aggravate damage of motoneurons in ALS [71]. Increased asynchronous release was observed in the neuromuscular junctions in mice lacking amyloid precursor protein, whose incorrect processing has the crucial role in pathogenesis of Alzheimer disease [72]. The processes of synchronization of ACh quanta release from mice motor-nerve terminals can be disrupted by poisoning with botulotoxin type D [73]. It has been established that the development of schizophrenia is accompanied by a reduced expression of *DTNBP1* encoding dysbindin protein. The decreased abundance of dysbindin causes a desynchronization of neurotransmitter secretion, which reduces the amplitude of evoked postsynaptic response and increases its duration [74]. High asynchrony of neurotransmitter secretion is seen in patients with familial hemiplegia migraine associated with the mutations in P/Q type Ca^2+^ channels, which is involved in the development of muscle weakness [75].

Taken together, these data give a reason to believe that searching for methods that can affect the time course of phasic synchronous release can lead to the creation of new therapies aimed at overcoming the consequences of synaptic defects [76].

## Figures and Tables

**Figure 1 biomedicines-10-01771-f001:**
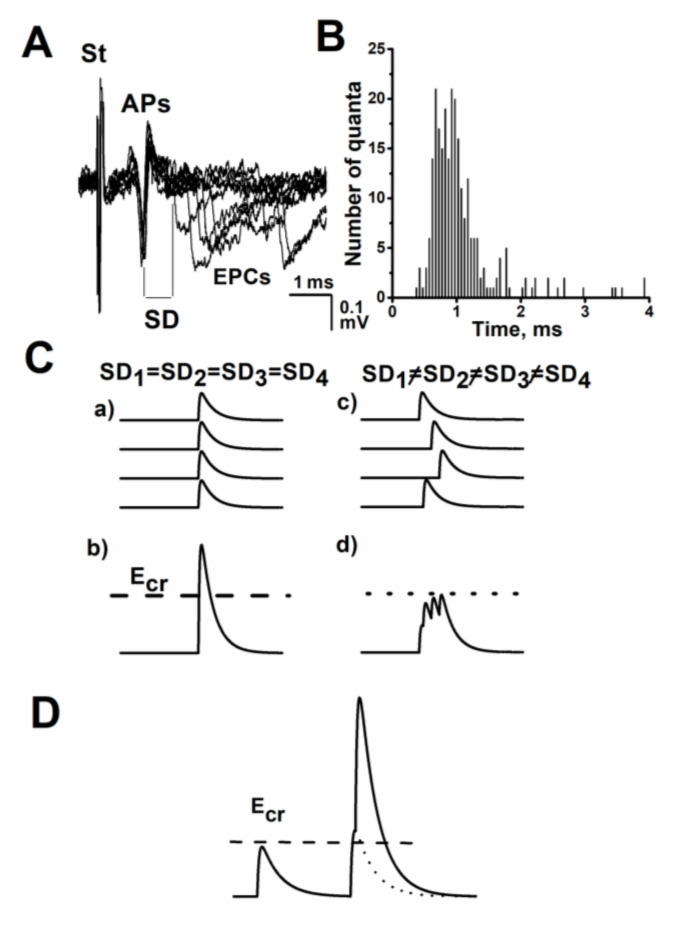
Asynchronous evoked phasic ACh release at the frog neuromuscular junction under low extracellular Ca^2+^ concentration and scheme demonstrating the role of asynchronous quantal release after a nerve stimulus in the formation of a multiquantal postsynaptic response. (**A**) Superimposed extracellular recordings of action potentials of the nerve terminal and uniquantal end-plate potentials under a low extracellular Ca^2+^ concentration. St—nerve stimulus; Aps—nerve action potentials; EPCs—end-plate currents; SD—real synaptic delay. (**B**) Histogram illustrating the distribution of synaptic delays of the uniquantal EPCs (data from single experiment). The bin size was 0.05 ms. (**C**) Summation of the uniquantal responses with equal time moments of release (a) and (b) as well as with different time moments of release (c) and (d); dash line and Ecr—the threshold for generation of muscle action potential, i.e., E critical level. (**D**) Scheme of generation of muscle action potential when the end-plate amplitude reaches the threshold for generation of the muscle action potential. This figure was prepared based on data from [7].

**Figure 2 biomedicines-10-01771-f002:**
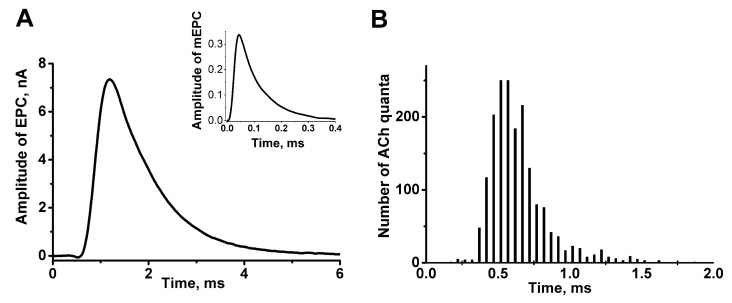
Asynchronous evoked phasic ACh release at extracellular concentration of Ca^2+^ close to the physiological level in the frog neuromuscular junction. (**A**) Averaged evoked multiquantal EPC recorded intracellularly under a physiological extracellular Ca^2+^ concentration, in the inset—averaged miniature EPC registered in the same neuromuscular junction; (**B**) the distribution of latent periods obtained by the subtraction from 256 multiquantal EPC curve, the average miniature EPC curve registered at the same neuromuscular junction. The ordinate shows the number of latent periods; the abscissa shows time in msec. This figure was based on [16].

**Figure 3 biomedicines-10-01771-f003:**
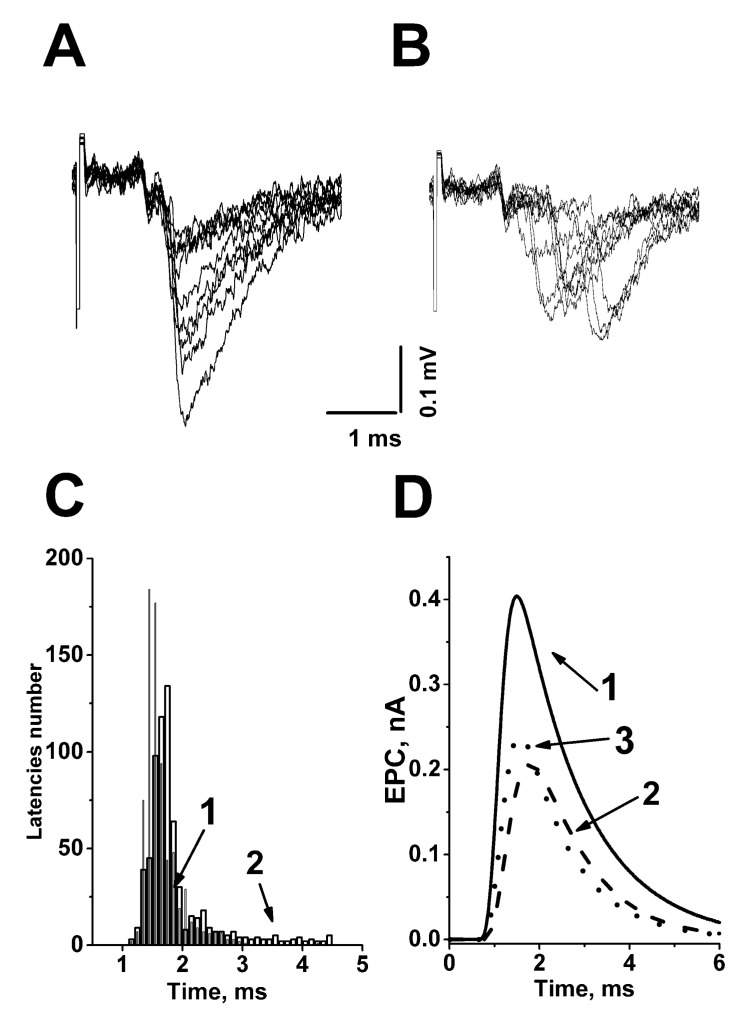
Exogenous ACh desynchronizes the quantal secretion in the frog neuromuscular junction. (**A**) Superimposed extracellular recordings of action potentials of the nerve terminal and uniquantal end-plate potentials under low extracellular Ca^2+^ concentration in control in the distal part of long nerve terminal; (**B**) the superposition of uniquantal EPCs in the same synapse after ACh (10 µM) addition in the bathing solution; (**C**) histogram illustrating the distribution of synaptic delays in control (1) and upon ACh action (2); (**D**) EPCs recorded in the control conditions (1), in response to 10 µM ACh (2). Reconstructed EPC without taking into account changes in the kinetics of quanta secretion is also shown (3). Note the latter response is larger by 17% than in the curve (2). This figure was based on [34,35,36].

**Table 1 biomedicines-10-01771-t001:** Effects of agonists and antagonists of AChRs on the time course of the evoked ACh quanta release.

Substance	Object	Receptor’s Type Action	Effects	References
ACh	frog	Agonist of nAChR and mAChR	Desynchronization (increase in synaptic delay dispersion)	[34]
Carbacholine	frog	Agonist of nAChR and mAChR		[35]
d-Tubocurarine	frog	Antagonist of nAChR		[29]
Nicotine	frog	Agonist of nAChR		[35]
Methoctramine	frog	antagonist of muscarinic M2/M4 mAChR	Desynchronization (slowed exponential decay on the synaptic delay histograms; slowed rise time of the multiquantal postsynaptic current)	[42]
Muscarine	frog	Agonist of mAChR	Synchronization (removing quanta with long synaptic delay) without preliminary treatment and after desynchronizing action of M2/M4 antagonists	[41,42]
Oxotremorine, Propargyl ester of arecaidine	Frog	M1 and M2 mAChR agonists	No effects	[35]
Pirenzepine, AF-DX 116, Methoctramine, VU 0255035, PD 102807, 4-DAMP and J 104129	frog	Antagonists of M1, M2, M3, M4 mAChRs		[35,41]
AF-DX 116	frog	M2 mAChR antagonist	Synchronization (shortening of the time interval during which 85% quanta are released)	[41]
Genetic blockade of muscarinic receptors	mouse	Knockout mice lacking functional M2 mAChRs	Desynchronization (longer duration of release in comparison to the control mouse)	[43]

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
