# Peer review of "Presynaptic Acetylcholine Receptors Modulate the Time Course of Action Potential-Evoked Acetylcholine Quanta Secretion at Neuromuscular Junctions"

_biomedicines, 2022, doi:10.3390/biomedicines10081771_

Round 1
Reviewer 1 Report
The review article by Bukharaeva et al. is dedicated to published studies describing pre-synaptic regulation of acetylcholine release. The text represents a good review of the literature from mid XX century to the latest years.
The manuscript has some minor issues:
line 31 "...process is called as a phasic..."
line 33 and throughout the manuscript: authors use term "real synaptic delay" without proper introduction. It does not easily deduced from the references that illustrate the respective paragraph. I suggest to disclose properly this term to avoid the possible reader confusion.
line 47 "real synaptic delay" again
line 189 "real synaptic delay" again
Table 1 in oxotremorine line: "Not effects" consider instead "No effects" or "not detected"
line 230 "As is known" sounds weird. It might be better with "It is known that"
line 242 the term "mobile Ca2+ buffer" is unclear. I suggest to express "mobility" in terms of capacity if possible;
line 242 "buffer will be is added"
lines 229, 252, 289 contain numbered list of propositions, but inside the second item of the list there used the same numbering type (i, ii, iii) as in the main list which is hard to understand. I suggest to change internal numbering to "a, b, c" or similar;
line 337 "asynchronous released";
line 376 "synaptic trans mission";
line 383 "trans mitter"
Reviewer 2 Report
The authors have discussed presynaptic acetylcholine receptors' role in releasing acetylcholine in the neuromuscular junction. The review is well written and mostly complete. My comments are provided below:
The role of many presynaptic receptors that modulate neurotransmitter release could be more clearly discussed. The cross-talk between receptors represents an important mechanism of neurotransmission modulation and plasticity. Perhaps it is worth illustrating it with a schematic figure.
Line 38, please discuss the relevance of this study in frogs to mice/humans.
Reviewer 3 Report
Bukharaeva and coworkers want to give their contribution doing a review about presunapltic acetylcholine receptors and their characteristic of modulate the time course of action potential-evoked acetylcholine quanta secretion at neuromuscular junctions. Their paper is well written and achieves the set goal.
